# Child outcomes after induction of labour or expectant management in women with preterm prelabour rupture of membranes between 34 and 37 weeks of gestation: study protocol of the PPROMEXIL Follow-up trial. A long-term follow-up study of the randomised controlled trials PPROMEXIL and PPROMEXIL-2

Annemijn A de Ruigh ,[1] Noor E Simons ,[1] Janneke Van 't Hooft,[1] Aleid G van Wassenaer-Leemhuis,[2] Cornelieke S H Aarnoudse-Moens,[2] Madelon van Wely,[3] Gert-Jan van Baaren,[1] Floortje Vlemmix,[1] D P van der Ham,[4] Augustinus S P van Teeffelen,[5] Ben W Mol ,[6] Tessa J Roseboom,[1,7] Eva Pajkrt[1]

For numbered affiliations see end of article.

**Correspondence to**
Dr Annemijn A de Ruigh;
a.a.deruigh@amsterdamumc.nl

## ABSTRACT

**Introduction** Late preterm prelabour rupture of membranes (PROM between $34^{+0}$ and $36^{+6}$ weeks gestational age) is an important clinical dilemma. Previously, two large Dutch randomised controlled trials (RCTs) compared induction of labour (IoL) to expectant management (EM). Both trials showed that early delivery does not reduce the risk of neonatal sepsis as compared with EM, although prematurity-related risks might increase. An extensive, structured long-term follow-up of these children has never been performed.

**Methods and analysis** The PPROMEXIL Follow-up trial (NL6623 (NTR6953)) aims to assess long-term childhood outcomes of the PPROMEXIL (ISRCTN29313500) and PPROMEXIL-2 trial (ISRCTN05689407), two multicentre RCTs using the same protocol, conducted between 2007 and 2010 evaluating IoL versus EM in women with late preterm PROM. The PPROMEXIL Follow-up will analyse children of mothers with a singleton pregnancy (PPROMEXIL trial n=520, PPROMEXIL-2 trial n=191, total IoL n=359; total EM n=352). At 10–12 years of age all surviving children will be invited for a neurodevelopmental assessment using the Wechsler Intelligence Scale for Children-V, Color-Word Interference Test and the Movement Assessment Battery for Children-2. Parents will be asked to fill out questionnaires assessing behaviour, motor function, sensory processing, respiratory problems, general health and need for healthcare services. Teachers will fill out the Teacher Report Form and answer questions regarding school attainment. For all tests means with SDs will be compared, as well as predefined cut-off scores for abnormal outcome. Sensitivity analyses consisting of different imputation techniques will be used to deal with lost to follow-up.

### Strengths and limitations of this study

► This long-term follow-up study will be the first study to evaluate long-term developmental outcomes (cognitive, motor, and behavioural development, sensory processing, respiratory problems, general health, children's need for healthcare services and school attainment) in the offspring of women who have been treated during the pregnancy with induction of labour or expectant management for late preterm prelabour rupture of membranes.

► Children will be evaluated at 10–12 years of age with internationally validated neurodevelopmental tests by a trained team consisting of a (neuro)psychologist and physician masked to the study group, and with questionnaires, translated for Dutch children, using norm scores for Dutch children.

► The study will be performed within the Dutch Consortium for Healthcare Evaluation and Research in Obstetrics and Gynaecology-NVOG Consortium 2.0, a collaboration of approximately 70 obstetric hospitals (academic and non-academic hospitals) in the Netherlands (NVOG; the Netherlands Society of Obstetrics and Gynaecology).

► Alongside this long-term follow-up study, a separately reported economic evaluation study will be planned to investigate cost-effectiveness of both treatments taking long-term developmental outcomes into account.

► The main limitation is that we expect to have an incomplete follow-up rate due to a high loss to follow-up, which we estimate to be 60%–70%. Baseline characteristics of children participating in follow-up versus lost to follow-up will be compared, to assess whether selection or attrition bias may be present in our study.

**Ethics and dissemination** The study has been granted approval by the Medical Centre Amsterdam (MEC) of the AmsterdamUMC (MEC2016_217). Results will be disseminated through peer-reviewed journals and summaries shared with stakeholders. This protocol is published before analysis of the results.

**Trial registration number** NL6623 (NTR6953).

## INTRODUCTION
### Background and rationale

Late preterm prelabour rupture of membranes (late preterm PROM) between $34^{+0}$ and $36^{+6}$ weeks gestation is an important clinical problem occurring in 1.5% of pregnant women, of which 25% will deliver within 24 hours.[1] After PROM, the risk of infection increases for both mother and fetus. Recently, three large randomised controlled trials (RCTs) compared induction of labour to expectant management for women whose pregnancy was complicated by late preterm PROM.[2–4] The Dutch PPRO-MEXIL and PPROMEXIL-2 trial (Expectant Management versus Induction of Labor (PPROMEXIL) and Expectant Management versus Inductionof Labor-2 Trial (PPRO-MEXIL-2)), and the Australian PPROMT (Preterm Pre-labour Rupture of Membranes close to Term Trial) showed that induction of labour does not reduce the risk of neonatal sepsis as compared with expectant management, while increasing prematurity related risks, such as hypoglycaemia and hyperbilirubinaemia. Furthermore, an individual participant data meta-analysis investigating participant data of all three RCTs also concluded that expectant management is an acceptable alternative to induction of labour, as both treatments resulted in comparable rates of a composite of adverse neonatal outcomes.[5] Moreover, an economic analysis of the PPRO-MEXIL trial showed that healthcare costs for induction of labour are slightly higher, although not statistically significant, with a mean difference of €754 (€8094 for induction of labour vs €7340 for expectant management, 95% CI: €335 to €1802).[6] Therefore, currently most national guidelines advocate expectant management for late preterm PROM.[1 7 8]

In 2015, our research team performed a follow-up study of children at 2 years of age, born to women who participated in the PPROMEXIL trial.[9] This follow-up study was performed with limited budget and used internationally validated screening questionnaires. Even though this study had a follow-up rate of 44% and no extensive neurodevelopmental assessments were used, an increase in neurodevelopmental impairment was found in the expectant management group as compared with the induction of labour group (abnormal score (−2 SD) in ≥1 domains of the Ages and Stages Questionnaire: 14% induction of labour group vs 26% expectant management group, difference in percentage −11.4; 95% CI −21.9 to −0.98).[9] Hypothetically, a prolonged stay of the fetus in an environment at risk for (subclinical) infections such as maternal placental inflammation (histological or clinical chorioamnionitis) and fetal side placental inflammation (funisitis and chorionic plate vasculitis) in case of

expectant management could affect brain development (ie, neurological outcome), and therefore, explain the neurodevelopmental impairment seen at 2 years of age.[10] The developmental effects of induction of labour or expectant management after late preterm PROM in children after 2 years are still unknown. Furthermore, understanding the long-term effects on women's offspring of either treatment is important for both clinicians and pregnant women when deciding how to manage late preterm PROM.

Until now, no other study has performed or planned a comprehensive long-term follow-up of children born after late preterm PROM. Study feasibility was investigated by an online questionnaire filled out by parents and members of a Dutch patient organisation representing patients affected by preterm birth due to complications in pregnancy. Results showed that 89% of parents were willing to participate in an extensive follow-up study. Parents rated the outcomes general health, behaviour, school attainment and respiratory problems as most important outcomes (data not published). A systematic review on neurodevelopment in preterm children showed a strong relationship between gestational age at delivery and cognitive abilities (ie, academic attainment, emotional and behavioural needs) in very, moderately and late preterm infants. These deficits persist beyond primary school for all neurodevelopmental domains. They stress the importance of knowledge on these long-term domains and advise trials to plan long-term follow-up to gain insight on possible neurodevelopmental delay in children.[11]

### Objectives

Therefore, the aim of this study is to conduct a structured follow-up of all children born to women with late preterm PROM who were randomised to induction of labour or expectant management in the PPROMEXIL and PPROMEXL-2 trial. Long-term cognitive, motor and behavioural development, sensory processing, respiratory problems, general health, children's need for healthcare services, and school attainment will be assessed at 10–12 years of age using internationally validated measurements and questionnaires, translated and using norm scores for Dutch children.

## METHODS AND ANALYSIS
### Study setting

We will perform an extensive long-term follow-up study of two previously executed RCTs (PPROMEXIL Follow-up trial, NTR 6953, METC 2016_217, NL58494.018.16) investigating long-term developmental outcomes (cognitive, motor, behavioural development), sensory processing, respiratory problems, general health, children's need for healthcare services, and school attainment. This will be assessed at 10–12 years of corrected age in children born to women with a singleton pregnancy complicated by late preterm PROM (between 34+0 and 36+6

de Ruigh AA, *et al. BMJ Open* 2021;**11**:e046046. doi:10.1136/bmjopen-2020-046046

weeks gestation), who participated in the RCTs PPRO-MEXIL and PPROMEXIL-2 trial. Details of the PPRO-MEXIL (ISRCTN29313500) and PPROMEXIL-2 trial (amendment of the PPROMEXIL trial (Medical Centre Amsterdam, MEC 05–240, ISRCTN05689407) have been published elsewhere.[2 3] These two large RCTs, using the same study protocol and conducted between 2007 and 2011 in 61 academic and non-academic hospitals in The Netherlands, assessed whether induction of labour vs expectant management would reduce the incidence of neonatal sepsis in women with late preterm PROM. In the induction of labour group, patients were induced within 24 hours after randomisation. Patients in the expectant management group were monitored until the onset of spontaneous delivery or induced after 37+0 weeks according to national guidelines.[1]

### Participants and eligibility criteria

All children born to women with a singleton pregnancy who participated in the PPROMEXIL trials will be invited for this long-term follow-up assessment. Children will be evaluated at 10–12 years of age. As the total number of multiple pregnancies in the PPROMEXIL and PPRO-MEXIL-2 trials was very low (14/727 (1.9%) and equally distributed among treatment groups), only singleton pregnancies will be included in the analysis (see figure 1).

### Procedures and recruitment

The study protocol is designed, constructed and reported according to the recommendations given in the Standard Protocol Items Recommendations for Interventional Trials (SPIRIT): SPIRIT checklist for reporting randomised trials (see online supplemental additional file 1, online supplemental additional file 2), SPIRIT schematic diagram of enrolment of PPROMEXIL follow-up participants, and the Guidance for Reporting Involvement of Patients and the Public (GRIPP2) short form (online supplemental additional file 3).[12] The study will be performed within the Dutch Consortium for Healthcare Evaluation and Research in Obstetrics and Gynaecology-NVOG Consortium 2.0, a collaboration of approximately 70 obstetric hospitals (academic and

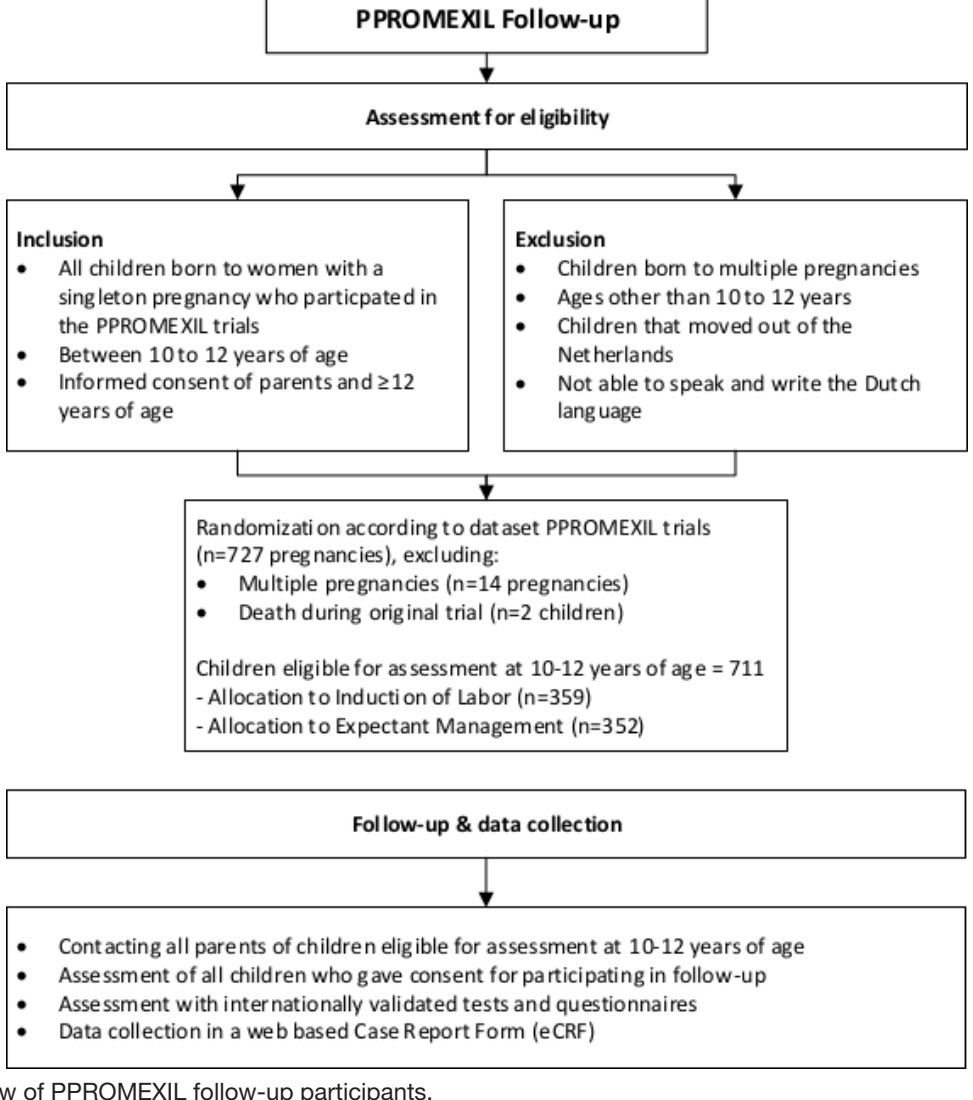

**Figure 1** Overview of PPROMEXIL follow-up participants.

non-academic hospitals) in the Netherlands (https://zorgevaluatienederland.nl/nvog) (NVOG; the Netherlands Society of Obstetrics and Gynaecology). Research nurses will be asked to crosscheck medical records for possible occurrence of death of women's offspring before contacting parents and their child for participating in this follow-up study. All parents will be contacted by post to announce this follow-up study, and if they give consent to be approached by the research team, they will be contacted by telephone or email to explain study details. Parents will be informed that participation is voluntary and that they may withdraw consent to participate at any time (see online supplemental additional file 4). They will be informed that declining participation will not affect their or their child's care. Parents will be given sufficient time to read the patient information and they will be given the opportunity to ask questions by telephone or email prior to signing the informed consent form. Written study information at children's reading level will be available for all children (specified for children <12 years of age and ≥12 years of age. An independent physician (ie, not a member of the research team) will be available to answer any questions patients may have. Written informed consent will be obtained from both parents prior to the examination. Children ≥12 years of age have to sign their own informed consent, in addition to the informed consent of their parents, at the day of the assessment. A copy of the informed consent form(s) will be given to the parents/child. All study documents will be available through the study website.

Concealment of treatment allocation at time of the PPROMEXIL and PPROMEXIL-2 trials (ie, induction of labour or expectant management) was not possible due to the type of intervention, and therefore, parents and children entered in this follow-up study will be aware of treatment allocation. The research team performing the follow-up examinations and all members of the research team performing data entry and data analyses will be masked to treatment allocation.

All data will be collected, captured and coded in accordance with existing polices to ensure patient confidentiality. Data will be recorded using an electronic case record form and will be stored in a web-secured database (available through the study website).[13] The investigators will publish the results of this trial in a peer reviewed medical journal as soon as appropriate. The clinical research unit of the Amsterdam University Medical Center (UMC) will monitor data collection.

### Follow-up assessment and outcomes

During a single visit in an outpatient clinic of a local hospital close to the family's neighbourhood, children will be assessed on long-term neurodevelopmental outcomes using standardised and validated neurodevelopmental tests and questionnaires. A trained team consisting of a (neuro)psychologist and physician, masked to the study group, will perform all neurodevelopmental tests. Neurodevelopmental assessment of children has a structured approach, is enjoyable for most children and is not invasive. During neurodevelopmental assessment of the child, parents will be asked to fill out questionnaires on sensory processing, behaviour, respiratory problems and child's health. If necessary, parents will be assisted with filling out questionnaires. All, but one, questionnaires are digital and can be filled out on a tablet during the assessment or at any other time at home.

After completing all examinations, parents receive a short report on their child's test results. This short report will give information on total test scores and tell parents whether their child's scores are above, on or below average. If test scores indicate that children would benefit from supportive (health, developmental or educational) care, parents are advised to contact their general practitioner for referral to a paediatrician or psychologist.

### Main study outcomes

► Cognitive development (Wechsler Intelligence Scale for Children -V (WISC-V)).
► Motor skills (Movement Assessment Battery for Children-2 (Movement-ABC-2)).
► Behaviour (Child Behaviour Checklist (CBCL)).

### Secondary study outcomes

► School attainment (Teacher's Report Form (TRF) and additional questions).
► Sensory processing (Short Sensory Profile-Netherlands (Short Sensory Profile-NL).
► Respiratory problems (International Study of Asthma and Allergies in Childhood questionnaire (ISAAC)).
► Pubertal status (Puberty Developmental Scale).
► General health (questionnaire).

### Assessment of cognitive development

Cognitive Development will be assessment using the Dutch version of the WISC-V.[14] The WISC-V is used worldwide to assess cognition in children aged 6–16 years and consists of 10 subtests that are combined into a Full Scale IQ score and five primary indexes: verbal comprehension, visual spatial, fluid reasoning, working memory and processing speed. Besides these primary indexes, an additional mathematics subtest will be obtained to provide an objective measurement of this area of academic attainment. The WISC-V total IQ score and primary indexes have a mean score of 100 points with an SD of 15 points. We will compare mean (SD) between treatment groups. Furthermore, an index score ≤70 (≥ −2 SD below the mean score) will be considered as a severe cognitive delay and will be compared between groups. An index score >70 and ≤84 (≥ −1 SD and < −2 SD below the mean score) will indicate a mild cognitive delay. Normal cognitive outcome is defined as no severe or mild neurodevelopmental delay. A difference between the two treatment groups of 7.5 points (0.5 SD) could indicate a potential clinical relevant difference.

Child's executive functioning will be tested using subtests of the WISC-V and the Color-Word Interference

Test (CWIT). The CWIT measures cognitive set shifting and the ability to inhibit a dominant and automatic verbal response by separate and combined Color Naming and Color Reading items. The CWIT subtests have a mean of 10 points with an SD of 2 points. An CWIT index score of ≤4 (ie, more than −2 SD below the mean score) is considered a severe delay in executive functioning and will be analysed.

## Assessment of motor skills

Child's motor function will be measured by the Movement Assessment Battery for Children-2 (M-ABC-2).[15] The M-ABC-2 is the most commonly used tool used to examine fine and gross motor skills. The M-ABC-2 provides data about a child's performance of age-appropriate tasks within three domains; manual dexterity, aiming and catching, and balance. M-ABC-2 scores will be calculated as standard scores and percentiles for each domain, and as a total test score. The mean standard score for all domains and the total score is 10 points, with an SD of 3 points. We will compare mean (SD) between treatment groups. The age band 2 (7–10 years of age) and 3 (11–16 years of age) of the M-ABC-2 will be used, as appropriate according to the child's age. Percentiles as defined by the M-ABC-2 testing manual and used in daily practice for testing motor skills in children will be applied. In short, a standard score of ≤5 points, representing ≤5th percentile will be defined as a significant movement difficulty and a severe delay in motor skills and will be compared between treatment groups. A standard score of 6 or 7 points, representing >5th to ≤16th percentile will indicate that the child is at risk of having a movement difficulty and therefore will be classified as mild delay in motor skills. A standard score of ≥8 points, representing >16th percentile will be defined as no movement difficulty and normal development of motor skills.

Additionally, parents will fill out the M-ABC-2 checklist, a questionnaire that consists of three sections on movement in static and/or predictable environment, movement in a dynamic and/or unpredictable environment and non-motor factors that may affect the child's movement. The sections on static and dynamic movements are summed up to a total score, with a higher score indicating a worse motor function. A total score of ≥95th percentile (≥9 points) indicates severe motor impairment and will be compared between both treatment groups.

## Assessment of behavioural development

Child's behaviour will be measured by the CBCL, a parental questionnaire used to screen for behaviour problems in children.[16] It informs on eight narrow syndrome scales (anxious/depressed, withdrawn/depressed, somatic complaints, social problems, thought problems, attention problems, rule-breaking behaviour and aggressive behaviour) and three broadband scales (internalising, externalising behavioural problems and a total problems score) which are composed out of the narrow-band syndrome scales. The CBCL broadband scales T scores have a mean of 50 points with an SD of 10 points. We will compare mean (SD) between treatment groups. Furthermore, a score >90th percentile (>63 points) on one of the two broad dimensions scales (internalising problems or externalising problems), or the total problem score (sum of all scores) of the CBCL will be defined as abnormal and clinically relevant for indicating behavioural problems. Scores ≥84th and ≤90th percentile (≥60 and ≤63 points) are considered borderline and scores <84th percentile (<60 points) are defined as normal.

## Assessment of school attainment

Child's academic attainment and behaviour will be assessed using the TRF.[17] The TRF assesses problem behaviour in the last 2 months and identifies the same eight syndromes as the CBCL, and also inquires on academic attainment (Academic Performance). With parental permission, the TRF will be filled out by the child's school teacher (the teacher who has known the child in the school setting for more than 2 months can complete the TRF). Accompanying the TRF, teachers will be asked some additional questions regarding the child's need for additional educational support inside or outside the classroom. For the TRF the cut-off percentiles of the broad band and total scores as used in the CBCL will be applied. For Academic Performance a cut-off score of <10th percentile (≤36 points) will be defined as abnormal. Scores between 10th and 16th percentile are classified as borderline and ≥17th percentile are considered normal outcome.

## Assessment of sensory processing

Sensory processing will be determined using the Short Sensory Profile questionnaire.[18] The Short Sensory Profile contains sections corresponding to each sensory system, sections that indicate the modulation of sensory input across sensory systems, and sections that indicate behavioural and emotional responses that are associated with sensory processing. This questionnaire consists of 38 items, classified into 7 subscales (Tactile Sensitivity, Taste/Smell Sensitivity, Movement Sensitivity, Underresponsive/Seeks Sensation, Auditory Filtering, Low Energy/Weak and Visual/Auditory Sensitivity). For every subscale, parents will be asked how frequently their children respond in the way described by each item using a 5-point Likert scale (nearly never, seldom, occasionally, frequently, almost always). Lower scores on the total score and subscales indicate more sensory symptoms. Subscales and the total scores will be used to classify as 'definite difference' (cut-off scores ≥−2 SD below the mean) and will be compared between groups. 'Typical performance' will be defined as < −1 SD below the mean, 'probable difference' will be defined as ≥ −1 SD and < −2 SD below the mean.

## Assessment of respiratory problems

Respiratory problems, such as asthma or other lung problems, will be assessed using the ISAAC questionnaire which informs on asthma, rhinitis and eczema.[19] The

diagnosis of asthma will be defined as a positive answers to the question: 'In the last 12 months, has your child had wheezing?', as this question has a sensitivity of 100%, specificity of 78%, positive predictive value of 73%, and negative predictive value of 100% for the diagnosis of asthma.[20]

### Assessment of anthropometry and pubertal status

Children will be asked to fill out the Puberty Developmental Scale (PDS), a self-report measure of pubertal status.[21] Children will be asked questions regarding on for example, growth in height, skin changes, body or facial hair, deepening of the voice (for boys), and starting to menstruate or developing breasts (for girls). Physical examination will be restricted to measurement of height/weight and blood pressure. Results of physical examination (height/weight, body mass index) will be used for baseline characteristics. Puberty status will be used for baseline characteristics and subgroup analysis.

### Assessment of child's health and need for healthcare services

A general questionnaire consisting of 61 items, will be used to assess demographic characteristics and will ask questions regarding the present (last 12 months) and the past health and healthcare use (from discharge after delivery until date of assessment) (also used in previous follow-up studies such as ProTWIN*kids* study at 3 and 4 years, TripleP study[22–24]). Questions address child's health, need for healthcare services, hospital visits, hospital submission, need for surgery, use of medication, psychological problems, need for developmental therapies (such as physical therapists, remedial teaching, speech therapist, occupational therapist). Healthcare use and (health) related problems will be clustered into different clinically relevant groups (eg, need for medical specialist and/or developmental care, medication use in the past and present, hospital admissions and surgery to give insight in the range of health-related problems).

Parents will be asked to give permission to gather medical information on the child's health via the general practitioner and the preventive youth healthcare services if needed.

### Economic analysis

Alongside this long-term follow-up study, an economic evaluation study will be planned to investigate cost-effectiveness of both treatments taking long-term developmental outcomes into account. Results of this economic evaluation will be reported separately from trial results.

At present, no additional long-term follow-up in later life (>12 years of age) is planned. Permission to approach parents and children for additional follow-up research in later life will be obtained with informed consent form during the current follow-up study. If additional long-term follow-up of children at an adolescence age will be planned in the future, additional approval of the Medical Research Ethics Committee will be sought.

### Sample size

Since this is a follow-up study, the maximum number of study participants is already defined by the two PPRO-MEXIL trials, excluding multiple pregnancies and deceased children (figure 1 and online supplemental additional file 2). Consequently, 711 children are eligible for inclusion, 359 born in the induction of labour group and 352 born in the expectant management group (PPROMEXIL trial n=520, PPROMEXIL-2 trial n=191). As we will not be able to adjust the number of recruited children, a power calculation will not be of any use to calculate a study sample size. However, this calculation can indicate the minimum number of children that need to be tested in order to find a clinically significant difference for the three main study outcomes: cognitive development, motor skills and behavioural development. All sample size calculations are with a power of 90%, a two-sided α of 0.05 and ß of 0.20. To be able to detect a clinically relevant difference in mean scores of 0.5 SD in the main outcomes, minimally 86 children per group are needed (total 172 children). This 0.5 SD equals a difference of 7.5 IQ points in the mean score of the WISC-V test (cognitive development), a difference of 1.5 points on the mean total standard scores of the M-ABC-2 (motor skills) and a difference of 5 points on the mean T scores in any of the broadband problem scales of the CBCL (behavioural development) between both groups. Thus, since 172 children comprise 24% of our total, also in case of limited follow-up, differences of 0.5 SD can be picked up. Based on previous experience in our research team with follow-up trials and based of existing literature, we expect to have a follow-up rate of 30%–40% of the children.[25]

### Statistical methods

Differences in background characteristics and the maternal, pregnancy, delivery and neonatal outcomes between the induction of labour group and expectant management group will be compared using unpaired t-test, Mann-Whitney U test, $\chi^2$ test or Fisher's exact test when appropriate. The same characteristics will be compared in children assessed at follow-up and for the original participants of the PPROMEXIL trials. This will allow us to assess whether selection or attrition bias may be present in our study (eg, due to drop-out of healthy or unhealthy children). To compare the long-term developmental outcomes between both treatment groups mean differences and the corresponding 95% CI will be calculated. For dichotomous outcomes relative risk (RR) with corresponding 95% CI will be calculated. Our main analyses shall be based on the results from the children assessed in follow-up (complete case analysis).[25]

The relatively simple statistical analysis described above can be justified by the fact that our study is a follow-up of two RCTs and consequently no confounding measures are expectant (see online supplemental additional file 5). The Direct Acyclic Graph confirms that there are no variables susceptive to have influenced the likelihood of

**Table 1** Chronology submission and revisions PPROMEXIL Follow-up study

| Version no. | Date (DD-MM-YYYY) | Main reasons for change |
|---|---|---|
| 1 | 29-09-2016 | N/A, first submission to MEC |
| 2 | 20-12-2017 | MEC2016_217#C20161752<br>Modifications requested by MEC d.d. 05-09-2016 |
| 3 | 04-01-2018 | Modifications requested by MEC d.d. 04-01-2018<br>Additional information on informed consent about saving data (mother and child) up to 15 years after trial |
| 3 | 10-01-2018 | Approval MEC d.d. 10-01-2018 |
| 4 | 31-05-2018 | Amendment 1:<br>► Administrative modifications.<br>► Change of acronym to PPROMEXIL Follow-up.<br>► Addition of two questionnaires (ISAAC and Puberty Developmental Scale). |
| 4 | 04-07-2018 | Approval amendment version 4 d.d. 04-07-2018 |
| 5 | 10-08-2018 | Amendment 2:<br>► Modification in protocol on how to recruit participants (via research nurses and PhD student).<br>► Modification in Patient Information Files on recruitment. |
| 5 | 23-08-2018 | Approval amendment version 5 d.d. 23-08-2018 |
| 6 | 06-11-2018 | Amendment 3:<br>► Clarification on informed consent procedure, parents and participants will be counselled through the telephone and sign informed consent at home.<br>► Minor modifications in the General Health Questionnaire. |
| 6 | 23-11-2018 | Approval amendment version 6 d.d. 23-11-2018 |
| 7 | 03-04-2019 | Amendment 4:<br>► Addition of patient information files for children age 12–15. |
| 7 | 12-04-2019 | Approval amendment version 7 d.d. 12-04-2019 |

ISAAC, International Study of Asthma and Allergies in Childhood questionnaire.

receiving the intervention and subsequently have influenced long-term outcome of the child. On the other hand, selection bias may occur as a consequence of incomplete follow-up. We will evaluate the effect of differences in background characteristics (such as maternal smoking, socialeconomic status) and if applicable, report unadjusted and adjusted ORs for dichotomous outcomes using logistic models and adjusted mean differences and the corresponding 95% CI for continuous outcomes using general linear models.

### Sensitivity analyses
Our pre-planned sensitivity analyses will only be performed for the WISC-V, the Movement-ABC and the CBCL total scores to minimise the effect of multiple testing.

#### Imputation missing data
A sensitivity analysis using imputation techniques will be performed to impute missing data for children that are lost to follow-up. Imputation techniques will only be applied when it can be assumed that data is (mostly) missing at random and the follow-up rate is follow-up rate ≥70% (the group agreed on an arbitrary). If the loss to follow-up rate is higher a best-case and worst-case scenario will be performed. In these two scenarios the missing cases are imputed either all as 'normal' (best case) or as 'abnormal' (worst case) outcomes. These scenarios will provide some insight on the robustness of the complete case follow-up results.

#### Age and puberty adjusted scores
Despite the fact that most children are born late premature/near term or full term, a sensitivity analyses will be performed using age-adjusted scores (corrected for prematurity). Finally, a sensitivity analysis using results of the PDS indicating child's puberty status will be performed.

### Subgroup analyses exploring the potential impact of effect modification
#### Dealing with the effect of 'down-stream' factors
During time to follow-up (due to loss to follow-up) a substantial difference in the prevalence 'down-stream' factors could potentially appear ('down-stream' factors are defined as potential effect modifiers appearing after randomisation, such as sepsis at birth, positive group B streptococci (GBS)). In sensitivity analysis the potential interaction of the following 'down-stream' factors will be explored: gestational age at PROM, receiving antibiotics, receiving steroids for fetal maturation, receiving tocolysis, GBS positivity, a positive vaginal culture (including GBS and other pathogens not consistent with normal flora),

**Table 2** WHO trial registration data set

| Primary registry and trial identifying number | Trial NL6623 (NTR6953) |
|---|---|
| Date of registration in primary registry | 28 December 2017 |
| Secondary identifying numbers | n/a |
| Source(s) of monetary or material support | ZonMW Dutch Healthcare efficacy programme |
| Primary sponsor | Academical Medical Centre, Amsterdam, The Netherland |
| Secondary sponsor(s) | n/a |
| Contact for public queries | Drs. Noor Simons<br>Follow-up.ppromexil@amsterdamumc.nl |
| Contact for scientific queries | Prof. dr. Eva Pajkrt<br>e.pajkrt@amsterdamumc.nl |
| Public title | PPROMEXIL Follow-up |
| Scientific title | Child outcomes after induction of labour or expectant management in women with preterm prelabour rupture of membranes between 34 and 37 weeks of gestation: the PPROMEXIL Follow-up trial, a long-term follow-up study of the randomised controlled trials PPROMEXIL and PPROMEXIL-2. |
| Countries of recruitment | The Netherlands |
| Health condition(s) or problem(s) studied | Late preterm prelabour rupture of membranes (PROM between $34^{+0}$ and $36^{+6}$ weeks gestational age). Long-term effects of induction of labour versus expected management. |
| Intervention(s) | n/a |
| Key inclusion and exclusion criteria | The PPROMEXIL Follow-up trial will analyse children of mothers with a singleton pregnancy (induction of labour n=359; expectant management n=352). At 10–12 years of (corrected) age all surviving children will be invited for follow-up. |
| Study type | Follow-up of a randomised controlled trial |
| Date of first enrolment | 3 August 2018 |
| Sample size | All sample size calculations are with a power of 90%, a two-sided α of 0.05 and ß of 0.20. To be able to detect a clinically relevant difference in mean scores of 0.5 SD in all tests, 86 children per group will be sufficient (total 172 children). This 0.5 SD equals a difference of 7.5 IQ points in the mean score of the WISC-V test (cognitive development), a difference of 1.5 points on the mean total standard scores of the M-ABC-2 (motor skills) and a difference of 5 points on the mean T scores in any of the broadband problem scales of the Child Behaviour Checklist, CBCL (behavioural development) between both groups. Thus, since 172 children comprise 24% of our total, also in case of limited follow-up, differences of 0.5 SD can be picked up. |
| Recruitment status | Open for patient inclusion |
| Primary outcome(s) | Cognitive development (WISC-V)<br>Motor skills (Movement-ABC-2)<br>Behaviour (CBCL). |
| Key secondary outcomes | Academic attainment and behaviour (Teacher Report Form)<br>Sensory processing (Short Sensory Profile)<br>Respiratory problems (International Study of Asthma and Allergies in Childhood questionnaire)<br>Pubertal status (Puberty Developmental Scale)<br>Height, weight, bloodpressure<br>General health and demographics (questionnaires) |
| Ethics review | The Medical Ethics Committee of the Academic Medical Centre Amsterdamhas approved the PPROMEXIL Follow-up trial (METC 2016_217, NL58494.018.16). |
| Completion date | n/a |
| Summary results | n/a |
| IPD sharing statement | n/a |

IPD, individual participant data; n/a, not available; WISC-V, Wechsler Intelligence Scale for Children.

de Ruigh AA, *et al. BMJ Open* 2021;**11**:e046046. doi:10.1136/bmjopen-2020-046046

neonatal sepsis and for women who participated in the former follow-up study of children at 2 years of age.[9] The analysis will be stratified by these different factors and the potential differences in long-term outcomes between the different strata will be explored.

A p<0.05 was considered to indicate statistical significance. All analyses will be performed according to the intention-to-treat principle using IBM SPSS Statistics 26 or in RStudio (Boston, Massachusetts, USA).

A statistical analysis plan, reporting a more detailed description of the statistical methods and analyses, will be published separately from the PPROMEXIL Follow-up protocol.

## Patient and public involvement

The Dutch association for parents of incubator children and the Dutch Collaboration of parent and patient organisations endorsed the study and provided input on the study proposal. Parents from the Dutch association for parents of incubator children participated in an online survey. Additionally, mothers of prematurely born children participated in a focus group meeting organised by our research team, to discuss the different aspects of child's long-term development to incorporate in long-term follow-up research.

## DISCUSSION

Long-term follow-up of all children born to mothers participating in obstetric intervention trials is of crucial importance.[26] The outcome late neurodevelopmental morbidity has been identified and selected by parents as one of 13 core outcomes for studies evaluating preventive interventions for preterm birth.[27] Furthermore, previous studies have stressed the importance of long-term follow-up by demonstrating that interventions performed during pregnancy can have unexpected long-term effects on children which may not be apparent at birth or during neonatal assessment.[28] By assessing cognition, motor function, behaviour, respiratory problems, general health and school attainment in an extensive and structured follow-up, this study will have the unique opportunity to help understanding the long-term effects of our current treatment regimen for late preterm PROM on women's offspring. Results from our study should be validated in other follow-up studies comparing induction of labour to expectant management.

## ETHICS AND DISSEMINATION
### Ethics approval and consent to participate

The PPROMEXIL Follow-up trial aims to assess long-term childhood outcomes of the PPROMEXIL trial (ISRCTN29313500) and PPROMEXIL-2 trial (MEC 05–240, ISRCTN05689407), two multicentre RCTs using the same study protocol. The Medical Ethics Committee of the Academic MEC has approved the PPROMEXIL Follow-up trial (MEC2016_217, NL58494.018.16). Table 1

describes the chronology of submission and amendments to the MEC.

See https://www.zorgevaluatienederland.nl/evaluations/ppromexil-follow-up for the full study protocol and electronic case record form. Written informed consent will be obtained from both parents prior to the examination. Children ≥12 years of age have to sign their own informed consent, in addition to the informed consent of their parents, at the day of the assessment. A copy of the informed consent form(s) will be given to the parents/child.

## Dissemination

No arrangements have been made concerning public disclosure. The trial is registered in the Dutch Trial register (Trial registration number: NTR6953. Date of registration 28 December 2017). An overview of the WHO trial registration data set is described in table 2.

Trial results will be submitted to a peer-reviewed journal, regardless of the outcome and made open access available in accordance with the Netherlands Organisation for Health Research and Innovation (ZonMW) policy. Results will be incorporated in national guidelines and patient information leaflets. Coauthorship will be based on the international committee of medical journal editor's guidelines. Contributors that not fulfil these criteria will be listed as collaborators. The order of authors will be based on scientific input.

## Availability of data and materials

The datasets used and/or analysed during the current-study will be available from the corresponding author on reasonable request.

**Author affiliations**
[1] Department of Obstetrics and Gynaecology, Amsterdam Reproduction & Development, Amsterdam UMC Location AMC, Amsterdam, The Netherlands
[2] Department of Neonatology and Paediatrics, Emma Children's Hospital, Amsterdam Reproduction & Development, Amsterdam UMC Location AMC, Amsterdam, The Netherlands
[3] Netherlands Satellite of the Cochrane Gynaecology and Fertility Group, Amsterdam University Medical Centres, Amsterdam, The Netherlands
[4] Department of Obstetrics and Gynaecology, Martini Hospital, Groningen, The Netherlands
[5] Department of Obstetrics and Gynaecology, Maastricht UMC+, Maastricht, The Netherlands
[6] Department of Obstetrics and Gynaecology, Monash University, Clayton, Victoria, Australia
[7] Department of Clinical Epidemiology, Biostatistics and Bioinformatics, Amsterdam UMC Locatie AMC, Amsterdam, The Netherlands

**Acknowledgements** We would like to thank the research nurses and research midwives from the Dutch Consortium for Healthcare Evaluation and Research in Obstetrics and Gynecology- NVOG Consortium 2.0 as well as the Dutch Consortium Trialbureau for their efforts. The authors thank the contributors of PPROMEXIL and PPROMEXIL-2 trial. For a list of contributors to the PPROMEXIL and PPROMEXIL-2 trials, see online supplemental additional file 6.

**Contributors** AAdR, NES, JVtH, AGvW-L, CSHA-M, MvW, G-JvB, FV, DPvdH, ASPvT, BWM, TJR, EP are member of the PPROMEXIL Follow-up trial study group and were involved in conception and design of the study. AAdR, NES, JVtH drafted the manuscript which follows the SPIRIT checklist for reporting randomised trials. AAdR, NES, JVtH, AGvW-L, CSHA-M, MvW, G-JvB, FV, DPvdH, ASPvT, TJR, BWM,

EP discussed and fine-tuned the final design of the study. All authors edited the manuscript and read and approved the final version of the manuscript.

**Funding** The study group received funding by ZonMW, the Netherlands Organisation for Health Research and Development (governmental funding), grant number: 843002826. ZonMW peer reviewed the primary study protocol, they had no other involvement in study design. ZonMw will not have any involvement in data collection, nor in analysis or writing of the manuscript.

**Disclaimer** Dr Ben Willem Mol is supported by a NHMRC Practitioner Fellowship (GNT1082548). Dr. Ben Willem Mol reports consultancy for ObsEva, Merck Merck KGaA and Guerbet. All other authors did not report any conflicts of interest.

**Competing interests** BWM is supported by an NHMRC Practitioner Fellowship (GNT1082548). BWM reports consultancy for ObsEva, Merck Merck KGaA and Guerbet. All other authors did not report any conflicts of interest.

**Patient consent for publication** Not required.

**Provenance and peer review** Not commissioned; externally peer reviewed.

**ORCID iDs**
Annemijn A de Ruigh http://orcid.org/0000-0003-2164-8917
Noor E Simons http://orcid.org/0000-0002-9109-0800
Ben W Mol http://orcid.org/0000-0001-8337-550X

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
