## [Reviewer comments · BMJ Open]

ARTICLE DETAILS

TITLE (PROVISIONAL)	Child outcomes after induction of labour or expectant management in women with preterm prelabour rupture of membranes between 34-37 weeks of gestation: Study protocol of the PPROMEXIL Follow-up trial A long-term follow-up study of the randomised controlled trials PPROMEXIL and PPROMEXIL-2.
AUTHORS	de Ruigh, Annemijn; Simons, Noor; Van 't Hooft, Janneke; van Wassenaer-Leemhuis, Aleid; Aarnoudse-Moens, Cornелиеke; van Wely, Madelon; van Baaren, Gert-Jan; Vlemmix, Floortje; van der Ham, D.P.; van Teeffelen, Augustinus; Mol, Ben; Roseboom, Tessa; Pajkrt, Eva

VERSION 1 – REVIEW

REVIEWER	Triebwasser, Jourdan University of Pennsylvania Perelman School of Medicine, Obstetrics & Gynecology
REVIEW RETURNED	27-Dec-2020

GENERAL COMMENTS	Is the abstract accurate, balanced and complete? - Recommend delineating how many participants from each trial will be included in the abstract. 6. Are the outcomes clearly defined? - It is unclear to me how 16th%ile was chosen as the upper bound of borderline test results.- Sample size calculated based on difference in mean score, while descriptions of the test focus on percentile scores. I would recommend aligning the description of the tests with intended analysis (mean SD vs. % with abnormal testing).- It is unclear what the primary outcome is. Having a subheading for "Study Outcomes" to clarify primary outcome would be very helpful.- There are many scores described that are not listed in sample size calculation. These should explicitly be described as secondary outcomes or exploratory analyses. 7. If statistics are used are they appropriate and described fully? - It seems there are a lot of tests occurring in a relatively small anticipated sample size. $p < 0.05$ without any correction could lead to several false positive findings.- If only 30-40% are anticipated to participate, is the imputation analysis reasonable? Additional comments: Background: - Given that the focus group data is not published, I think this paragraph could be omitted. There may be published data about
---

	important outcomes for parents of preterm babies that could instead be cited. Or the findings at 2 year seem enough rationale to have a larger, longer-term follow-up. - FOODNOTES--is this supposed to say "Footnotes"? - Additional figure file 4--I don't find this graphic very helpful. We know events in the 10-12 years after randomization may have a profound effect on developmental scores, and that many of these factors are interrelated.
REVIEWER	Gallot, Denis Centre Hospitalier Universitaire de Clermont-Ferrand, obstetrics and gynaecology
REVIEW RETURNED	02-Jan-2021
GENERAL COMMENTS	This study protocol is highly well-written and of great interest. Based on 2 famous RCTs comparing active vs expectant management in case of preterm rupture of membranes between 34 and 36+6 GW, it aims to describe child outcomes at the age of 10-12 years. I have strictly no additional comment. Congratulations and waiting from future results and conclusion of this crucial study!

VERSION 1 – AUTHOR RESPONSE

Reviewer: 1

Dr. Jourdan Triebwasser, University of Pennsylvania Perelman School of Medicine Comments to the Author:

Is the abstract accurate, balanced and complete?

Question 1. Recommend delineating how many participants from each trial will be included in the abstract.

Response: We would like to thank the reviewer for this comment. The PPROMEXIL Follow-up trial will analyse children of mothers with a singleton pregnancy who originally participated in the PPROMEXIL (n=520) and PPROMEXIL-2 trials (n=191). We will exclude multiple pregnancies (n=14) and children who died during the original trial (n=2), leaving a total of n=712 children (induction of labour n=359; expectant management n=352) Please see Figure 1 and table 2.

Changes made: Line 83-84: "The PPROMEXIL Follow-up trial will analyse children of mothers with a singleton pregnancy (PPROMEXIL trial n= 520, PPROMEXIL-2 trial n=191, total induction of labour n=359; total expectant management n=352)."

and line 429-430: "Consequently, 711 children are eligible for inclusion, 359 born in the induction of labour group and 352 born in the expectant management group (PPROMEXIL trial n=520, PPROMEXIL-2 trial n=191)."

Question 2: It is unclear to me how 16th%ile was chosen as the upper bound of borderline test results.

Response: The 16th percentile mentioned in the manuscript represents a cut-off (percentile) score as defined by the Movement Assessment Battery for Children-2 (M-ABC-2). M-ABC-2 scores are calculated as standard scores and percentiles for three different domains (manual dexterity, aiming and catching, and balance). Based off the standard score and the associated percentile children can be classified as having a significant moment difficulty (severe delay: ≤5th), classified as being at risk

of having a movement difficulty (mild delay: >5th to ≤16th), or classified as having no movement difficulty (normal development: >16th percentile). These percentiles are defined by the M-ABC-2 testing manual and used in daily practice for testing motor skills in children.

Changes made: Line 326-328: "Percentiles as defined by the M-ABC-2 testing manual and used in daily practice for testing motor skills in children will be applied."

Question 3: Sample size calculated based on difference in mean score, while descriptions of the test focus on percentile scores. I would recommend aligning the description of the tests with intended analysis (mean SD vs. % with abnormal testing).

Response: For the three main study outcomes (WISC-V, M-ABC-2, CBCL) we have clarified that the primary analyses is the comparison of mean(SD) scores in the description of the test section. Additionally we also define the cut-off scores used to compare % abnormal testing.

Changes made: Line 304-305, line 324 and line 350: "We will compare mean (SD) between treatment groups. Furthermore, [...]"

Question 4: It is unclear what the primary outcome is. Having a subheading for "Study Outcomes" to clarify primary outcome would be very helpful.

Response: In the proposed follow-up study no primary outcome will be defined. We consider our (most important) main outcomes to be: cognitive development (assessed by the WISC-V), motor skills (assessed by the M-ABC-2) and behavioural development (assessed by the CBCL). We have clarified the difference between our main outcomes and secondary outcomes in the methods section.

Changes made: line 283-293

"Main study outcomes

- Cognitive development (Wechsler Intelligence Scale for Children -V)
- Motor skills (Movement-ABC-2)
- Behaviour (Child Behaviour Checklist)

Secondary study outcomes

- School attainment (Teacher's Report Form and additional questions)
- Sensory processing (Short Sensory Profile-NL)
- Respiratory problems (International Study of Asthma and Allergies in Childhood questionnaire)
- Pubertal status (Puberty Developmental Scale)
- General health (questionnaire)"

Question 5: There are many scores described that are not listed in sample size calculation. These should explicitly be described as secondary outcomes or exploratory analyses.

Response: We have clarified the difference between our main outcomes and secondary outcomes in the manuscript. This is in line with our response and changes made regarding question 4.

Changes made: Line 283-293:

"Main study outcomes

- Cognitive development (Wechsler Intelligence Scale for Children -V)
- Motor skills (Movement-ABC-2)
- Behaviour (Child Behaviour Checklist)

Secondary study outcomes

- School attainment (Teacher's Report Form and additional questions)
- Sensory processing (Short Sensory Profile-NL)

- Respiratory problems (International Study of Asthma and Allergies in Childhood questionnaire)
- Pubertal status (Puberty Developmental Scale)
- General health (questionnaire)"

Line 432-435: "However, this calculation can indicate the minimum number of children that need to be tested in order to find a clinically significant difference for the three main study outcomes: cognitive development, motor skills and behavioural development."

Question 6: It seems there are a lot of tests occurring in a relatively small anticipated sample size. $p < 0.05$ without any correction could lead to several false positive findings.

Response: We agree and consider our calculated sample size of 172 as a minimal sample size needed to be able to detect a difference of 0.5SD. With an alpha of 0.025 this would be a minimum of 202. Please also see our response to editorial question 1

Changes made: line 436-438: "To be able to detect a clinically relevant difference in mean scores of 0.5 SD in the main outcomes, minimally 86 children per group are needed (total 172 children)."

We added to our discussion, line 527-528: "Results from our study should be validated in other follow-up studies comparing induction of labour to expectant management."

Question 7: If only 30-40% are anticipated to participate, is the imputation analysis reasonable?

Response: We appreciate the question posed by the reviewer. In case of a high rate of cases loss to follow-up, imputation techniques become 'unstable'. For that reason, our main analysis will be based upon the results from all children assessed in follow-up (complete case analysis). Imputation techniques will solely be performed as sensitivity analysis. Furthermore, imputation techniques will only be applied when the follow-up rate is $\geq 70\%$. If the loss to follow-up rate is higher, a best- and worst-case scenario will be performed. This best- and worst-case scenarios will provide some insight on the robustness of the complete case follow-up results.

Please see line: 474-481

Changes made: none.

Additional comments:

Background:

Question 8: Given that the focus group data is not published, I think this paragraph could be omitted. There may be published data about important outcomes for parents of preterm babies that could instead be cited. Or the findings at 2 year seem enough rationale to have a larger, longer-term follow-up.

Response: We appreciate the question posed by the reviewer. Even though our focus group provided us very valuable information, we agree with the reviewer that the paragraph can be omitted, since the data is not published. We have deleted the information about the focus group and added information about long-term neurodevelopmental consequences of preterm birth.

Changes made: line 176-182: "A systematic review on neurodevelopment in preterm children showed a strong relationship between gestational age at delivery and cognitive abilities (i.e. academic attainment, emotional and behavioural needs) in very, moderately and late preterm infants. These deficits persist beyond primary school for all neurodevelopmental domains. They stress the importance of knowledge on these long-term domains and advise trials to plan long-term follow-up to

gain insight on possible neurodevelopmental delay in children.”

Question 9: FOODNOTES--is this supposed to say "Footnotes"?

Changes made: Line 573, Changed to “Footnotes”.

Question 10: Additional figure file 4--I don't find this graphic very helpful. We know events in the 10-12 years after randomization may have a profound effect on developmental scores, and that many of these factors are interrelated.

Response: We would like to thank the reviewer for this comment. It is known that confounding measures can influence results. However, we will be performing a long-term follow-up study of two RCTs, and therefore no confounding measures are expected. This can be confirmed (visually) by a Direct Acyclic Graph (DAG). As shown in Additional file 4, the DAG confirms that there are no variables susceptible to have influenced the likelihood of receiving the intervention and subsequently have influenced long-term outcome of the child.

On the other hand, it is known that parental socioeconomic disadvantages, independently from pregnancy and delivery complications, are associated with abnormal child neurodevelopment in early life. Furthermore, it is also known that incomplete follow-up can attribute to selection bias. Therefore, we will evaluate the effect of differences in background characteristics (such as maternal smoking, social-economic status) and if applicable, report unadjusted and adjusted odds ratios for dichotomous outcomes using logistic models and adjusted mean differences and the corresponding 95% CI for continuous outcomes using general linear models.

Changes made: None. We would like to keep Additional file 5 in the manuscript if possible, to justify for the relatively simple statistical analysis described in Line 460-464.

Reviewer: 2

Prof. Denis Gallot, Centre Hospitalier Universitaire de Clermont-Ferrand Comments to the Author: This study protocol is highly well-written and of great interest. Based on 2 famous RCTs comparing active vs expectant management in case of preterm rupture of membranes between 34 and 36+6 GW, it aims to describe child outcomes at the age of 10-12 years. I have strictly no additional comment. Congratulations and waiting from future results and conclusion of this crucial study!

Response: We would like to thank the reviewer for these compliments. As the reviewer stated, this long-term follow-up study will be the first study to evaluate long-term developmental outcomes (cognitive, motor, and behavioural development, sensory processing, respiratory problems, general health, children's need for health-care services, and school attainment) in the offspring of women who have been treated during pregnancy with induction of labour or expectant management for late preterm prelabour rupture of membranes. We believe that this study will have the unique opportunity to help understanding the long-term effects of our current treatment regimen for late preterm PROM on women's offspring.

Changes made: None.